# Antigenic Determinant of *Helicobacter pylori* FlaA for Developing Serological Diagnostic Methods in Children

**DOI:** 10.3390/pathogens11121544

**Published:** 2022-12-15

**Authors:** Hyun-Eui Park, Seorin Park, Damir Nizamutdinov, Ji-Hyeun Seo, Ji-Shook Park, Jin-Su Jun, Jeong-Ih Shin, Wongwarut Boonyanugomol, Jin-Sik Park, Min-Kyoung Shin, Seung-Chul Baik, Hee-Shang Youn, Myung-Je Cho, Hyung-Lyun Kang, Woo-Kon Lee, Myunghwan Jung

**Affiliations:** 1Department of Microbiology, College of Medicine, Gyeongsang National University, Jinju 52727, Republic of Korea; 2Institute of Health Science, Research Institute of Life Science, Gyeongsang National University, Jinju 52727, Republic of Korea; 3BK21 Center for Human Resource Development in the Bio-Health Industry, Department of Convergence Medical Science, Gyeongsang National University, Jinju 52727, Republic of Korea; 4Department of Pediatrics, College of Medicine, Gyeongsang National University, Jinju 52727, Republic of Korea; 5Department of Sciences and Liberal Arts, Amnatcharoen Campus, Mahidol University, Amnatcharoen 37000, Thailand; 6Research Institute of Life Science, Gyeongsang National University, Jinju 52828, Republic of Korea

**Keywords:** *Helicobacter pylori*, FlaA, ELISA, children

## Abstract

The early diagnosis of *Helicobacter pylori* infection is important for gastric cancer prevention and treatment. Although endoscopic biopsy is widely used for *H. pylori* diagnosis, an accurate biopsy cannot be performed until a lesion becomes clear, especially in pediatric patients. Therefore, it is necessary to develop convenient and accurate methods for early diagnosis. FlaA, an essential factor for *H. pylori* survival, shows high antigenicity and can be used as a diagnostic marker. We attempted to identify effective antigens containing epitopes of high diagnostic value in FlaA. Full-sized FlaA was divided into several fragments and cloned, and its antigenicity was investigated using Western blotting. The FlaA fragment of 1345–1395 bp had strong immunogenicity. ELISA was performed with serum samples from children by using the 1345–1395 bp recombinant antigen fragment. IgG reactivity showed 90.0% sensitivity and 90.5% specificity, and IgM reactivity showed 100% sensitivity and specificity. The FlaA fragment of 1345–1395 bp discovered in the present study has antigenicity and is of high value as a candidate antigen for serological diagnosis. The FlaA 1345–1395 bp epitope can be used as a diagnostic marker for *H. pylori* infection, thereby controlling various gastric diseases such as gastric cancer and peptic ulcers caused by *H. pylori*.

## 1. Introduction

*Helicobacter pylori* (*H. pylori*) is a pathogen that infects approximately 50% of the global population [1] and colonizes the epithelial surface or the surface mucus of the gastric mucosa [2]. It is known to increase the risk of gastric cancer by causing chronic inflammation and peptic ulcers [1,3,4,5,6]. Therefore, *H. pylori* was defined as a class 1 carcinogen by the World Health Organization (WHO) in 1994. According to a report released by the International Agency for Research on Cancer in 2020, *H. pylori* was responsible for the most infectious pathogen-related carcinogenesis in 2018 (https://gco.iarc.fr/causes/infections; accessed on 11 December 2022). Therefore, the early diagnosis of *H. pylori* infection is of great importance for the early diagnosis of gastric cancer. Early diagnosis is also important for developing a preventive method for peptic ulcers and gastric cancer based on the risk assessment induced by *H. pylori* infection [7,8]. In addition, given that most *H. pylori* infections are established in childhood [9,10,11,12], it is necessary to develop novel diagnostic methods applicable to pediatric patients.

Endoscopic biopsy is the most widely used method for diagnosing *H. pylori* infection [7,8]. The sensitivity of this method varies depending on the operator’s proficiency. In addition, not all *H. pylori* infections lead to gastric lesions, such as gastritis or peptic ulcers [5,13]; therefore, accurate biopsy is difficult. In pediatric patients, an endoscopic biopsy is difficult to perform. Moreover, in pediatric patients, an endoscopic biopsy is not appropriate as an early diagnostic method for *H. pylori* infection, considering that most *H. pylori* infections are established at a young age and there are numerous cases where gastric lesions are not observed, even after infection [5,8,10,11,13,14,15].

Serological diagnosis may be an alternative method for resolving these issues [14]. It could also be an appropriate tool for children in whom gastric lesions have not been observed. However, there is also a limitation in applying serological tools for diagnosing *H. pylori* infection, as the useful diagnostic antigens of *H. pylori* have not been identified [5,13,14,16,17]. Moreover, the highly heterogeneous characteristics of virulence factors known to be associated with host immune responses make it difficult to find useful antigen candidates for enzyme linked immunosorbent assay (ELISA) [4,13,17,18,19].

Thus, it is necessary to identify specific *H. pylori* antigens that show high immunogenicity to develop a more useful and convenient method for serological diagnosis. FlaA is a necessary protein for the survival of *H. pylori* in the gastric environment [17,20] and is also known to have high immunogenicity [21]. Therefore, this study investigated the effective antigens in the FlaA to develop an ELISA method for the early diagnosis of *H. pylori* infection as a novel diagnostic approach that can also be applied to pediatric patients.

## 2. Materials and Methods

### 2.1. Bacterial Strain

*Helicobacter pylori* strain 51 (*H. pylori* 51), isolated at Gyeongsang National University Hospital from a patient with a gastric disorder, was obtained from the *H. pylori* Korean Type Culture Collection (College of Medicine, Gyeongsang National University, Korea; 2006–2015). *H. pylori* 51 was grown on Brucella agar (Difco, Franklin Lakes, NJ, USA) containing 10% bovine serum (Difco) at 37 °C under 10% CO_2_ and 100% humidity [22,23]. All *Escherichia coli* strains for recombinant protein expression were cultured in Luria-Bertani (LB) broth (Invitrogen, Waltham, MA, USA) containing 1 mM ampicillin (Sigma, St. Louis, MO, USA) at 37 °C.

### 2.2. Serum Samples of Patients

Sera from pediatric patients with (*n* = 60) and without (*n* = 220) *H*. *pylori* infection were collected at Gyeongsang National University Hospital (GNUH, Jinju, Korea). Biopsy specimens were smeared onto Brucella agar plates containing 10% bovine serum, vancomycin (Sigma, 10 μg/mL), nalidixic acid (Sigma, 25 μg/mL), and amphotericin B (Sigma, 1 μg/mL) to culture *H. pylori* [22,23]. The bacteria were identified as *H. pylori* based on the 16s rRNA PCR using primers of PV31 (5′-CGGCCCAGACTCCTACGGG-3′) and PV32 (5′-TTACCGCGGCTGCTGGCAC-3′) [24]. A campylobacter-like organism (CLO) test was also conducted to measure urease activity using three biopsy samples from the same patient according to a previously described method [25]. Urease activity was determined over 24 h based on the time of color change of the broth. Sixty serum samples from patients with *H. pylori* infection were divided into two groups based on the results of the CLO test: weak (color change within 6–24 h, *n* = 30) and strong (color change within 6 h, *n* = 30). All sera were stored at −70 °C until further use. All biopsy and serum samples were collected under parental consent and provided by Gyeongsang National University Hospital, a member of the National Biobank of Korea, after obtaining permission from the hospital ethics committee (GNUHIRB-2016-04-003-001).

### 2.3. PCR Amplification of H. pylori FlaA Fragments and Cloning

Total genomic DNA was obtained from *H. pylori* 51 using the QiaAmp DNA Mini Kit, according to the manufacturer’s instructions (Qiagen, Hilden, Germany). According to the GenBank database, full-length *flaA* (1533 bp) was found to include *NdeI* (gagttacacat) and *BglII* (cccgggagat) in the N- and C-terminal regions, respectively (Appendix A). The primers for amplification of full-length *flaA* were designed based on published alignments available through GenBank (Table 1). The amplified full *flaA* gene was cloned into the pGEM-T vector. The pGEM-T vector was digested with *NdeI* and *BglII*, and the fragment of the full *flaA* gene was purified and retrieved from an agarose gel after electrophoresis. Purified *flaA* was cloned into the pET15b vector and treated with *NdeI* and *BglII* (pET15b/flaA). For the 1–772 bp fragment (Frag 1) of the full *flaA* gene, the pET15b/flaA vector was digested with *BamHI* and *BglII* and ligated (pET15b/Frag 1). For the flaA 778–1533 fragment (Frag 2), PCR was performed using the template of total genomic DNA of *H. pylori* 51, a forward primer with the *NdeI* recognition sequence and the *BamHI* recognition site, and a reverse primer used for amplification of the full *flaA* gene. The cloning process was performed in the same way as that for full *flaA* gene cloning.

Frag 2, which showed higher antigenicity than Frag 1, was divided into subfragments of 200–230 bp with 50% overlap. Fragments showing relatively high antigenicity were selected among subfragments of 200–230 bp and cloned as subfragments of 100 bp in size. The 100 bp subfragments showing the highest antigenicity were finally divided into 50 bp subfragments. Antigenicity analysis was performed using Western blotting. The primers used to prepare each subfragment are listed in Table 1. Primers for amplifying all subfragments except Frag 2–2 were designed to include *EcoRI* and *XhoI* recognition sequences in the F- and R-primers, respectively. The primers used for Frag 2–2 amplification were designed to contain *BamHI* and *XhoI* recognition sites in F- and R-primers, respectively.

All amplified subfragments were cloned into the pGEMT/A vector. These vectors were digested with *EcoRI*/*XhoI* or *BamHI/XhoI* and subsequently purified from an agarose gel after electrophoresis. The purified subfragments were cloned into the pEGexp vector and treated with *EcoRI, XhoI,* or *BamHI and XhoI*. The pEGexp vector was constructed based on the pET15b vector, which includes an N-terminal leader consisting of the amino acid sequence of GroEL T1 (1–772 bp of the *H. pylori GroEL* gene) and chaperonin [26]. GroEL T1 is highly expressed in *E. coli* and has a *HindIII* recognition sequence at the 771 bp position of the gene without any reactivity to the sera of *H. pylori*-infected patients (Figure 1). Thereafter, cloned genes were transformed into competent *E. coli* BL21 cells for expression [27].

The PCR was performed in a PCR PreMix tube (Accupower PCR PreMix, Bioneer, Daejeon, Korea) containing 250 µM dNTP, 1 U of Taq DNA polymerase, 10 mM Tris-HCl (pH 9.0), 40 mM KCl, 40 mM MgCl_2_, a stabilizer tracking dye, 40 ng of the template, and 2 pmol of primers, using a PCR Thermal Cycler (Takara, Kusatsu, Japan). PCR amplification was performed under the following conditions: 94 °C for 10 min; 35 cycles of 94 °C for 1 min, 50 °C for 1 min, and 72 °C for 2 min; and 72 °C for 10 min. The amplified PCR products were subjected to electrophoresis on a 1% agarose gel and purified using a QIAquick Gel Extraction Kit (Qiagen) [27]. All restriction enzymes were obtained from Roche Holding AG (Basel, Switzerland).

### 2.4. Expression and Purification of Recombinant Proteins

*E. coli* BL21 cells were grown in LB broth until the culture reached an optical density (OD600) of 0.6–0.8. Thereafter, isopropyl β-Ɗ-1-thiogalactopyranoside (1 mM; Duchefa Biochemie, Haariem, Netherlands) was added and subsequently incubated for 4 h. Thereafter, harvested cells were resuspended and lysed by sonication in a lysis buffer (50 mM Tris-hydrogen chloride, 100 mM sodium chloride, pH 8.0). The lysates were collected as supernatants by centrifugation at 5000× *g* for 30 min. The recombinant proteins were purified using His-Tag purification column, according to the manufacturer’s instructions (Bio-Rad, Hercules, CA, USA). All purified recombinant proteins were stored at –20 °C until use.

### 2.5. Immunoblot Analysis of the Recombinant FlaA Fragments

Purified recombinant proteins were separated using 12% sodium dodecyl sulfate-polyacrylamide gel electrophoresis (SDS-PAGE) and stained with Coomassie blue R-250 (Bio-Rad) or with Ponceau S (ThermoFisher, Waltham, MA, USA) after electrotransformation onto nitrocellulose membranes (Bio-Rad). For Western blot analysis, recombinant proteins were electrotransferred onto nitrocellulose membranes, which were immunoblotted with pooled 300 *H. pylori*-positive sera verified previously at a 1:200 dilution [28] and an alkaline phosphatase-conjugated goat anti-human IgG (H + L) at a 1:500 dilution (Bio-Rad). Specific reactions were detected using an AP-conjugated substrate kit (Bio-Rad). The reactivity of immunoblot was compared and graded using the mean gray value function of the ImageJ software (1.53e version) after background compensation. The relative antigenicity values are the ratio of Western blot reactivity to the protein staining levels in SDS-PAGE (loading control).

### 2.6. Enzyme Linked Immunosorbent Assay Using an Antigenic Determinant of FlaA

The determined antigen (10 µg/mL) was coated in 96-well microplates at 4 °C for 16 h for the ELISA. Following blocking at 37 °C for 3 h using 3% bovine serum albumin fraction V (BSA; Fisher Scientific, Hampton, NH, USA) in PBS containing 0.05% Tween 20 (PBST; Sigma), the plates were incubated with sera (1:100 diluted with 1% BSA in PBST) from patients at 37 °C for 1 h. Thereafter, they were incubated with horseradish peroxidase-conjugated goat anti-human IgG or IgM (1:10,000 diluted with 1% BSA in PBST; Bethyl, Montgomery, TX, USA) at 37 °C for 1 h. Between each step, plates were washed with PBST. Color development was performed by incubating the samples with o-phenylene diamine as a substrate and subsequently stopped with 2 N sulfuric acid, after which the microplates were read at 492 nm using an Emax Precision microplate reader (Molecular Devices, San Jose, CA, USA) [26].

### 2.7. Statistics

Data are presented as the mean ± standard deviation (SD). Statistical significance was determined with the Student’s *t*-test using Statistical Package for Social Sciences software (version 24, SPSS, Chicago, IL, USA). Differences were considered significant if *p* < 0.05.

## 3. Results

### 3.1. Antigenic Determination of H. pylori FlaA

The pooled sera from *H. pylori*-infected patients showed strong reactivity against the FlaA whole protein on Western blot analysis (Figure 2). The IgG in pooled sera was also bound to Frag 1 and 2; however, it showed a stronger reactivity to Frag 2 than to Frag 1. To effectively produce recombinant proteins and soluble subfragment proteins of Frag 2, they were cloned into the pEGexp vector. The resulting recombinant proteins were expressed and conjugated with the GroEL T1 protein, which is the N-terminal 771 bp portion of the *H. pylori* GroEL protein. On Western blot analysis using pooled sera from *H. pylori*-infected patients, the GroEL T1 protein did not show reactivity (Figure 1). Strong reactivity of pooled sera from *H. pylori*-infected patients was observed against 200–230 bp subfragments of Frag 2: Frag 2–5 (1195–1395 bp) and 2–6 (1297–1533 bp). Between Frag 2–5 and 2–6, a 100 bp subfragment that showed high reactivity with IgG in the sera was Frag 2–5–2 (1297–1395 bp). In Frag 2–5-2, the 50 bp subfragment that exhibited the highest reactivity was Frag 2–5–2–2 (1345 –1395 bp, ISTVNNISITQVNVKAA), which was selected as the effective antigen for ELISA (Table 2, Figure 3 and Figure 4, and Appendix A).

### 3.2. Analysis of the Reactivity of H. pylori-Infected Pediatric Patients Antibodies to the FlaA Fragment 2–5–2–2

The Frag 2–5–2–2 was purified and used as an ELISA antigen for the diagnosis of *H. pylori* infection. ELISA was performed with sera from *H. pylori*-infected patients (CLO weak group, *n* = 30; CLO strong group, *n* = 30) and non-infected children (CLO negative group, *n* = 220). The average absorbance of IgG ELISA was 0.25 ± 0.29, 0.93 ± 0.27, and 0.89 ± 0.33 in the sera of non-infected, CLO weak, and CLO strong groups at 492 nm, respectively (Figure 5, *p* < 0.000). In the case of IgM ELISA, the average absorbance was 0.09 ± 0.08, 1.04 ± 0.31, and 1.02 ± 0.37 in the sera of non-infected, CLO weak, and CLO strong groups, respectively (Figure 6, *p* < 0.000). No significant differences were observed between the weak and strong CLO groups. When the serum samples were divided into non-infected and *H. pylori*-infected (CLO positive) groups, the cut-off value of 0.53 for IgG ELISA was determined via receiver operating characteristic (ROC) analysis, with 90.0% sensitivity and 90.5% specificity, whereas the cut-off value of 0.35 for in IgM ELISA was determined via ROC analysis with 100% sensitivity and specificity.

## 4. Discussion

*H. pylori* is a highly heterogeneous bacterium with various virulence factor antigens that may be involved in the pathogenicity and immune responses [4,13,17,18]. Several virulence factors have been suggested as potential diagnostic antigens [17,29,30,31,32]. However, not all virulence factor antigens are present in all *H. pylori* strains. Indeed, the existence and expression of several virulence factors differ depending on the strain distribution in different areas. Several previous studies have reported geographical differences in the prevalence of *H*. *pylori* virulence factors [4,6,11,15,16,17,19,33,34,35,36,37]. The expression of these virulence factors can be regulated via their own modulatory system according to changes in environmental conditions [5,38,39,40]. These results imply that different types of antigens are exposed to the immune system of the host at various levels, depending on the environment in which *H. pylori* adapts. Moreover, not all virulence factor antigens induce an immune response following infection [13].

Flagellar antigens have a high possibility of being diagnostic antigens because the flagella play an important role in bacterial survival in the gastric environment [17,20]. After entering the stomach of the host, *H. pylori* must neutralize the hostile acidic conditions at the beginning of infection [41,42] and move towards the gastric epithelium of the host to establish successful colonization [17,20]. Urease and the flagella mediate acid neutralization and bacterial motility, respectively [17,20,41,42]. Of these, urease is an enzyme whose expression level may vary depending on the characteristics of the strain and environmental pH; however, as flagellar antigens are structural proteins, it is considered more valuable as diagnostic antigens [43,44]. Therefore, this study focused on the antigenicity of the flagella.

Flagellar antigens are essential structural proteins for the survival and colonization of *H. pylori*; therefore, they might be exposed to the immune system of the host continuously, thereby inducing specific immune responses including humoral immunity [17,45]. These results are consistent with those of a previous study demonstrating that mice immunized with flagellar sheath proteins exhibited significantly reduced *H. pylori* colonization [20]. Moreover, the high antigenicity of flagella-related proteins was reported in previous studies, which suggested that the flagella antigen is a suitable diagnostic and vaccine target [21,32,46]. The flagellar filament consists of two flagellins (FlaA and FlaB) encoded by *flaA* and *flaB*. Between FlaA and FlaB, FlaA has been suggested as a marker for the presence of *H. pylori* infection and gastric cancer [32,46]. Altogether, previously reported characteristics of FlaA show a high possibility of being a specific antigen required for the serological diagnosis of *H. pylori*. Therefore, in the current study, we identified a specific antigen in the FlaA protein and analyzed whether the identified antigen could be used in ELISA for the diagnosis of *H. pylori*.

There have been several attempts to use FlaA as a diagnostic antigen [32,46]. However, the sensitivity and specificity of the diagnostic methods depend on the ability of a specific antigen to induce the most specific and detectable immune responses [47,48]. Considering that the FlaA size is 56.7 KDa and that the size of the antigen in major histocompatibility complex (MHC) I or MHC II does not exceed 30 amino acids, the large size of FlaA (511 amino acids) should be fragmented to determine the specific antigen that has a high serological diagnostic value without a non-specific reaction. Therefore, in the current study, the *flaA* gene was divided into pieces, step-by-step, and each of them was cloned to produce recombinant proteins (Figure 4). While the recombinant proteins of full-sized FlaA, Frag 1, and 2 were expressed in the soluble form, the others were not. Therefore, other fragment antigens were expressed by binding them to the GroEL T1 protein of *H. pylori* to produce a soluble form [26]. The *H. pylori groEL* gene has a high expression level after transformation into *E. coli* and it has *a HindIII* restriction enzyme site at the 771 bp position, thereby easily removing the sequence below the 771 bp position. The recombinant protein of the N-terminal 1–772 bp of *H. pylori* GroEL (GroEL T1) did not show antigenicity on Western blotting using pooled sera from *H. pylori*-infected patients. All recombinant proteins conjugated with GroEL T1 were subjected to Western blotting using pooled sera of *H. pylori*-infected patients, thereby identifying the Frag 2–5-2–2 (1345–1395 bp, ISTVNNISITQVNVKAA) of having major antigenicity. As mentioned above, it is very important that the antigen used for diagnosis is a conserved antigen of *H. pylori*. Diagnostic methods using an antigen of a variable region have many disadvantages in their sensitivity and specificity. The Frag 2–5–2–2 was found to be a highly conserved antigen according to the NCBI database.

The recombinant protein Frag 2–5–2–2 showed specific immune responses in the sera of *H. pylori*-infected patients (CLO positive groups) as a diagnostic antigen in ELISA. The IgG ELISA OD values of the *H. pylori*-infected and non-infected groups were 0.91 ± 0.30 and 0.25 ± 0.29, respectively (Figure 5, *p* < 0.000). For IgM, OD values of 1.03 ± 0.34 and 0.09 ± 0.08 were observed in the *H. pylori*-infected and non-infected groups with a significant difference, respectively (Figure 6, *p* < 0.000). Moreover, ROC analysis showed a cut-off value of 0.53 for IgG ELISA with 90.0% sensitivity and 90.5% specificity, and a cut-off value of 0.35 for IgM ELISA, with 100% sensitivity and specificity. These results show that the determined antigen in the present study has high antigenicity and is an important candidate antigen for serological diagnosis, especially in pediatric patients. There were no significant differences in the OD values of IgG and IgM between the weak and strong CLO groups. It appears to be due to the different expression levels of ureA and ureB according to the environment of survival and colonization of *H. pylori* [43,44].

*H. pylori* infection increases the risk of gastric cancer, the fourth leading cause of cancer-related deaths worldwide. However, when gastric cancer is diagnosed early, the five-year relative survival rate increases [49,50]. Therefore, the early diagnosis of *H. pylori* infection can provide important clues for the prevention and treatment of gastric cancer. Moreover, *H. pylori* infection increases the risk of gastritis and gastric ulcers; therefore, early diagnostic tools for *H. pylori* infection with high sensitivity and specificity, before the development of clinical symptoms, are required to effectively overcome gastric diseases.

As major diagnostic methods, gastroscopy and biopsy have low sensitivity when considering that many cases of *H. pylori* infection are established in childhood without any clinical symptoms. Moreover, they are stressful for pediatric patients. As a non-invasive diagnostic method, the urea breath test and stool antigen test have also been used in clinical practice. The urea breath test is an exhalation investigation method to measure the hydrolysis of ^13^C or ^14^C isotope-labeled urea by *H. pylori* [51,52]. However, as the cutoff value of this method has not been determined, the results need to be validated differently considering the targeted population [51,53,54]. In addition, because the urea breath test requires expensive equipment, it is difficult to use widely due to its high cost [51]. The stool antigen test is a diagnostic tool based on *H. pylori* antigen detection in feces. It can be used for patients who cannot use the urea breath test, such as pediatric or asthmatic patients [55,56,57,58]. However, the accuracy of this diagnostic method can be affected by various gastrointestinal factors, such as bleeding ulcers, antibiotic treatment, and bismuth-containing compounds [57,59]. Therefore, it is necessary to develop a simple and accurate method for the early diagnosis of *H. pylori* infection that can be applied to pediatric patients.

The purpose of the present study was to discover a specific antigen of FlaA that can be used in ELISA for the serological diagnosis of *H. pylori* infection, which can be easily applied to all patients and is convenient for application in children. Previous studies showed different results regarding the sensitivity of the detection of IgG and IgA for *H. pylori* diagnosis [51,60,61,62,63]. Nevertheless, the measurement of IgG levels in sera can be said to be the gold standard for diagnosis [62,63,64]. Furthermore, considering the possible use of the FlaA Frag 2–5–2–2 as a diagnostic antigen for further multi-epitope ELISA, as indicated by the results of the current study, the IgG results will be more useful than those of IgA. Therefore, including IgM, our study focused on the IgG responses to the discovered recombinant antigens.

Among 220 sera samples in the non-infected group, 21 samples showed higher OD values of IgG ELISA than the cut-off value. Of these, ten samples showed values higher than 0.91 ± 0.30, the average IgG ELISA OD of sera samples from infected patients. Misdiagnosis of *H. pylori* infection is considered the primary reason for these high OD values measured in the sera of some patients in the non-infected group. The negative pediatric sera we used were collected from patients diagnosed to be negative for *H. pylori* through endoscopic biopsy. However, in some pediatric patients, *H. pylori* might not be accurately diagnosed. Therefore, even though they were positive, an endoscopic biopsy could be used to diagnose them as being negative for *H. pylori*. According to a previous study, the sensitivity of endoscopic biopsy in Korea has been estimated to be 69% [65]. It can be expected that true positive sera may be included in the non-infected group. For accurate identification of the reason, follow-up medical records are required. However, since there was no record of subsequent diagnosis, we could not confirm whether the patients of the non-infected group showing high OD levels in sera were subsequently diagnosed to be *H. pylori* positive. Another possibility is a non-specific reaction to Frag 2–5–2–2. Commonly, serological tests of ELISA cannot exclude non-specific responses completely. However, considering that the size of antigens associated with reactivity, except for the conjugated GroEL T1, is 50 bp, the possibility of a non-specific reaction seemed low. Moreover, most sera in the non-infected group showed lower OD values than the cut-off value. Therefore, it is believed that the misdiagnosis of *H. pylori* infection caused noise in the results by disturbing sample grouping.

The antigen of Frag 2–5–2–2 (1345–1395 bp, ISTVNNISITQVNVKAA) in FlaA was discovered in this study, and the ELISA results showed its potential as a diagnostic antigen. The Frag 2–5–2–2 can be used as an antigen in diagnostic kits for *H. pylori* infection, thereby helping control various gastric diseases caused by *H. pylori*. Further studies are needed to discover other effective antigens; thus, it will be possible to develop ELISA or immunochromatography kits with improved sensitivity and specificity based on the multi-epitopes.

## Figures and Tables

**Figure 1 pathogens-11-01544-f001:**
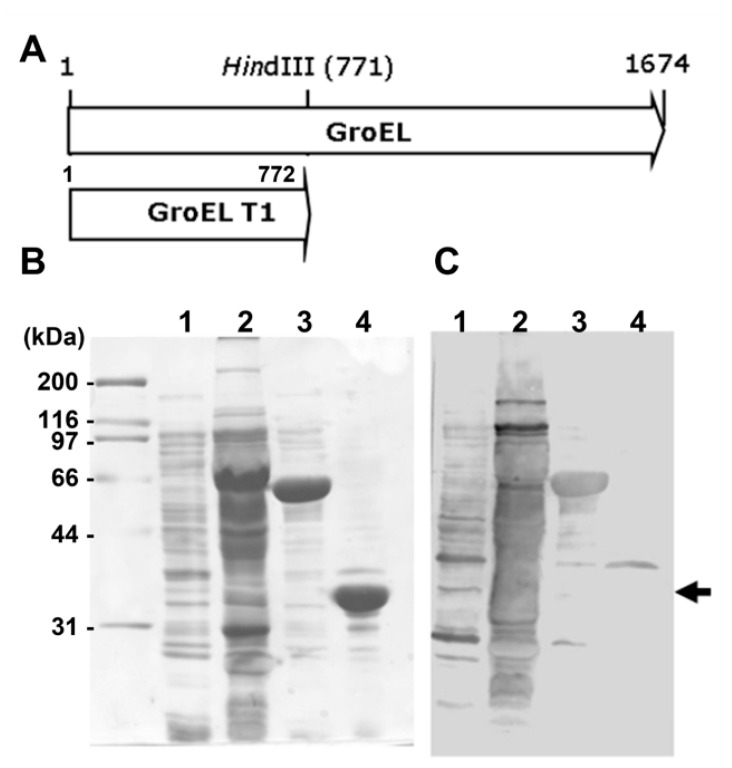
Immunoblot analysis of recombinant proteins of GroEL and GroEL T1. (**A**) The map of GroEL and GroEL T1 genes of *H. pylori*. (**B**) SDS-PAGE and (**C**) Western blot analysis using *H. pylori*-positive pooled sera. Black arrows indicate GroEL T1 (772 bp) recombinant protein, which did not show any reactivity to *H. pylori*-positive pooled sera. Line 1, whole lysate of *E. coli* containing pET15B; line 2, whole lysate of *H. pylori* 51; line 3, recombinant protein of GroEL; line 4, recombinant protein of GroEL T1.

**Figure 2 pathogens-11-01544-f002:**
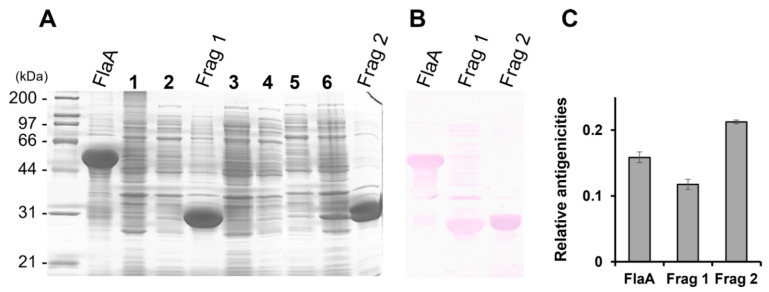
(**A**) SDS-PAGE and (**B**) Western blot analysis of the recombinant proteins, FlaA, Frag 1, and Frag 2. (**C**) Normalized relative antigenicity of each recombinant protein. All recombinant proteins showed reactivity to *H. pylori*-positive pooled sera. The positive pooled sera showed a stronger reactivity to Frag 2 than to Frag 1. 1 and 2, uninduced FlaA; 3 and 4, uninduced Frag 1; 5 and 6, uninduced Frag 2.

**Figure 3 pathogens-11-01544-f003:**
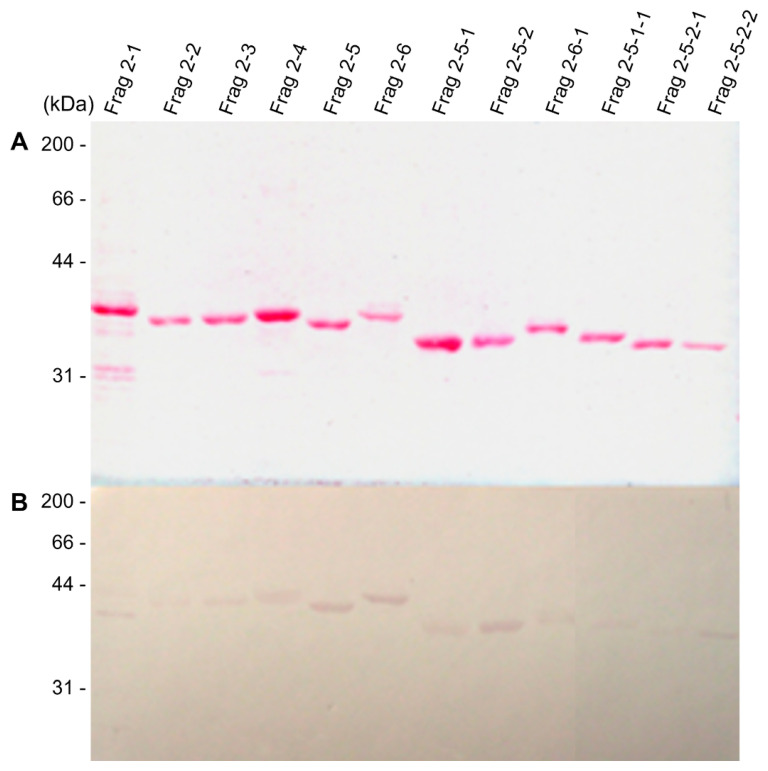
Immunoblot analysis of the recombinant subfragment proteins of FlaA. (**A**) Ponceau S staining and (**B**) Western blot analysis using *H. pylori*-positive pooled sera. High levels of relative reactivities were observed in Frag 2–5 (1195–1395 bp), Frag 2–6 (1297–1533 bp), Frag 2–5–2 (1297–1395 bp), and Frag 2–5–2–2 (1345–1395 bp).

**Figure 4 pathogens-11-01544-f004:**
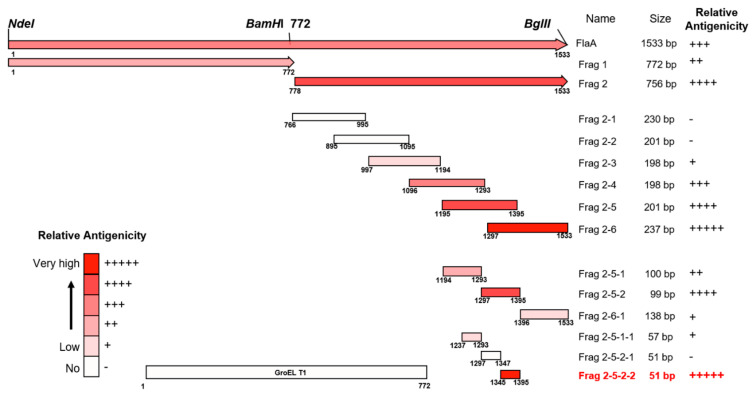
Maps of antigenic domains of FlaA. Except for Frag 1 and Frag 2, all subfragment proteins of FlaA were expressed by conjugation with GroEL T1 protein (772 bp).

**Figure 5 pathogens-11-01544-f005:**
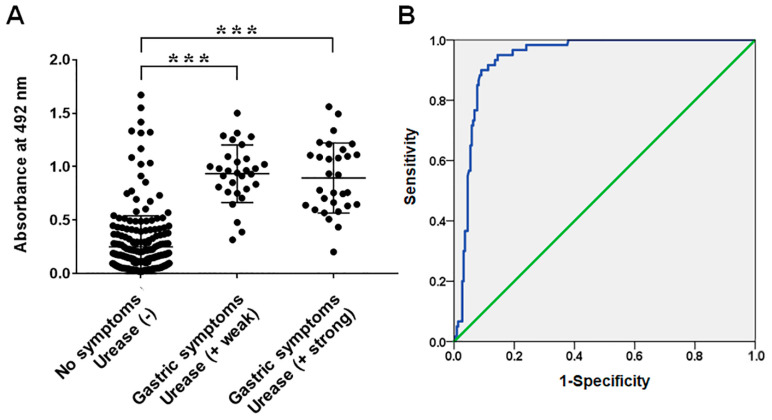
(**A**) IgG reactivities of sera from infant patients with/without *H. pylori* infection measured via ELISA using recombinant protein Frag 2–5-2–2 of FlaA. (**B**) Receiver operating characteristic (ROC) curves of ELISA using Frag 2–5-2–2 for the diagnosis of *H. pylori* infection. The area under the ROC of Frag 2–5-2–2 ELISA was 0.939 (95% confidence interval; *** *p* < 0.000).

**Figure 6 pathogens-11-01544-f006:**
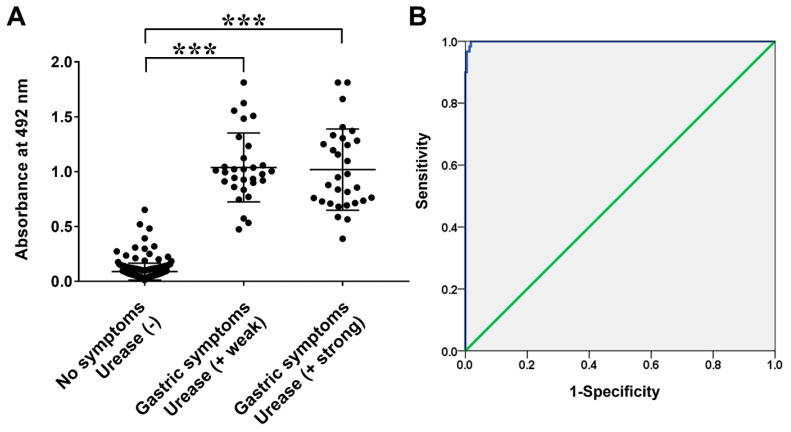
(**A**) IgM reactivities of sera from infant patients with/without *H. pylori* infection measured via ELISA using recombinant protein Frag 2–5-2–2 of FlaA. (**B**) Receiver operating characteristic (ROC) curves of ELISA using Frag 2–5-2–2 for the diagnosis of *H. pylori* infection. The area under the ROC of Frag 2–5-2–2 ELISA was 0.999 (95% confidence interval; *** *p* < 0.000).

**Table 1 pathogens-11-01544-t001:** Oligonucleotide primers for PCR amplification of flaA and its fragments.

Primer		Sequences (5′–3′)	Position (bp)	Length (aa)
FlaA (full)	F	gagttacacatATGGCTTTTCGGGTCAAT	1–1533	510
	R	cccgggagatCTAAGTTAAAAGCCTTAAGATATTTTGTTG		
Frag 1		Digestion of pET15b/flaA vector with *BamHI* and *BglII* and ligation		257
Frag 2	F	atgggatccatatgGGTAATATCGCAGATATTAAGAAAAA	778–1533	251
	R	cccgggagatCTAAGTTAAAAGCCTTAAGATATTTTGTTG		
Frag 2–1	F	cccccgaattcGGGATCCATTTGGGTAAT	766–995	76
	R	cccccctcgagTTTGTTAAATCCTGACCG		
Frag 2–2	F	ggcgcggatccTTGCGCAGTATAGATGGT	895–1095	67
	R	cccaaactcgagCGCTGTGAAACCTAAATG		
Frag 2–3	F	cccccgaattcGGCTCTACAAACTACGGA	997–1194	66
	R	acagcgctcgaGTTCGCGCCACTGGCTGA		
Frag 2–4	F	cccccgaattcATTGGTTTTGGGGAATCT	1096–1293	66
	R	atccaactcgagCATCGCAGACTCGGCAAT		
Frag 2–5	F	gtggcgaattcTATAACGCTGTCATCGC	1195–1395	67
	R	tttgagctcgAGCCGCTTTAACATTCAC		
Frag 2–6	F	ctgcggaattcATGTTGGATAAAGTCCGC	1297–1533	78
	R	cccccctcgagCTAAGTTAAAAGCCTTAA		
Frag 2–5–1	F	gtggcgaattCTATAACGCTGTCATCGC	1194–1293	33
	R	atccaactcgagCATCGCAGACTCGGCAAT		
Frag 2–5–2	F	ctgcggaattcATGTTGGATAAAGTCCGC	1297–1395	33
	R	tttgagctcgAGCCGCTTTAACATTCAC		
Frag 2–6–1	F	gggcccgaattcGAATCTCAAATTAGG	1396–1533	45
	R	cccccctcgagCTAAGTTAAAAGCCTTAA		
Frag 2–5–1–1	F	gggcccgaattcGGGGTTACAACCTTA	1237–1293	19
	R	atccaactcgagCATCGCAGACTCGGCAAT		
Frag 2–5–2–1	F	ctgcggaattcATGTTGGATAAAGTCCGC	1297–1347	17
	R	gggcccctcgagAATCATTTGATTTTG		
Frag 2–5–2–2	F	gggcccgaattcATTAGCACCGTGAAT	1345–1395	17
	R	tttgagctcgAGCCGCTTTAACATTCAC		

**Table 2 pathogens-11-01544-t002:** Measured relative antigenicity of the recombinant subfragment proteins of FlaA.

Frag No.	2–1	2–2	2–3	2–4	2–5	2–6	2–5-1	2–5-2	2–6-1	2–5-1–1	2–5-2–1	2–5-2–2
Relative antigenicity	0.051 ± 0.04	0.001 ± 0.05	0.077 ± 0.06	0.123 ± 0.08	0.210 ± 0.06	0.268 ± 0.03	0.102 ± 0.05	0.210 ± 0.05	0.070 ± 0.06	0.066 ± 0.06	0.017 ± 0.06	0.267 ± 0.08
Grade	-	-	+	+++	++++	+++++	++	++++	+	+	-	+++++

## Data Availability

The data supporting the conclusions of this article are provided within the article. The original data in the present study are available from the corresponding authors.

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
