# Peer review of "Antigenic Determinant of Helicobacter pylori FlaA for Developing Serological Diagnostic Methods in Children"

_pathogens, 2022, doi:10.3390/pathogens11121544_

Round 1

Reviewer 1 Report

General Comments

The manuscript by Park and colleagues entitled "Antigenic Determinant of Helicobacter pylori FlaA for Developing Serological Diagnostic Method in Children” describes a novel serological test for H. pylori based on a fragment of the FlaA antigen. The manuscript is clearly written and the techniques used for antigen isolation and detection are sound. The particular importance of this test method in children is also addressed, although comparison with currently available H. pylori testing requires further discussion.

Major Comments

1. The authors should cite a recent comprehensive review article about H. pylori (Ansari & Yamaoka, Clin Microbiol Rev. 2022;35:e0025821). The usual diagnostic methods for this pathogen are the urea breath test and stool antigen test, while serology has fallen out of favor for reasons discussed in the review article. The breath and stool tests are not particularly reliable due to medication interference, however, and a validated serological test seems preferable, especially in children. The authors should address these issues rather than rehashing the problems with endoscopic diagnosis in the Discussion section.

2. The authors have examined IgM and IgG serology with their FlaA fragment. Why was IgA not included? This marker may be very useful to demonstrate active H. pylori infection, especially in children (Kosunen et al, Lancet.1992; 339:893; Herbrink & van Doorn, Eur J Clin Microbiol Infect Dis. 2000;19:164).

3. One problem with serological testing for H. pylori is that antibodies can persist for some time after eradication of the organism. How does FlaA serology measure up in this regard?

4. A novel RecomLine H. pylori test system using multiple recombinant antigens has recently been described (Formichella et al, J Immunol Res. 2017;2017:8394593). How does the simpler FlaA test compare to this more complex analysis?

5. Although the authors state that using the full FlaA antigen is more cumbersome than the antigenic fragment, it appears that the whole antigen works well on Western blots (Figure 2). Would use of the whole antigen increase sensitivity of the test?

Minor Comments

1. The first sentence of the Introduction should be rewritten: "Helicobacter pylori (H. pylori) is a pathogen that infects approximately 50% of the global population and colonizes the epithelial surface or the surface mucus of the gastric mucosa [1]"  

2. Western blot should have a capital “W".

Reviewer 2 Report

This manuscript reported antigenic region of one of flagellin proteins, FlaA of Helicobacter pylori against serum IgG and IgM from Korean children. Authors prepared full and partial recombinant FlaA proteins in E. coli and purified. Series of sub-fragmented FlaA recombinant proteins were expressed as GroEL N-terminal fusion protein with N-terminus of GroEL from pET15b based pEGex vector. Purified GroEL-N half fused-FlaA segment proteins were used to determine antigenicity to pool of positive serum IgG. The most reactive smallest antigenic region of FlaA were Frag 2-5-2-2, located near C-teriminus of FlaA. This GroEL-N half fused-FlaA-Frag 2-5-2-2 protein were fixed on ELISA plate and used to detection of serum IgG and IgM from Korean children and were shown good performances.

              Authors worked well experimentally, however presentation of data are not well organized scientifically. Modification of manuscript are required.

Major point:

1.      Current Figure 1: Do SDS-PAGE of full FlaA, Fragment 1, and Fragment 2 in one gel again to show the image without figure cut and fusion as like current Figure 1A. One more SDS-PAGE with 1/10 amount of proteins is needed at the same time for the western blotting as Figure 1B to show the difference of detection of Fragment 1 and Fragment 2. Current Figure 1B are too much amount of recombinant protein to show the difference of pool-sera reactivity to Frag 2 more than Frag1. Detection of western botting should be shown more contrast in image at least by blue detection on membrane such as NBT/BCIP, or chemiluminescenece. In the materials and methods, authors described usage of ImageJ software, so show the quantification data of western blotting of figure 1B by a bar graphs as Figure 1D, normalized with amounts of recombinant proteins on membrane stained PonseauS (Please show as Figure 1C same as other figures). All the figures can be smaller than current figures.

2.      Current Figure 2 and Figure 3: Author should have enough recombinant proteins, so do again Ponceaus S staring of membrane and pool-sera western blotting to have clear image same as above, then quantification to show evidence of antigenicity in Figure 4. Both of figures can be smaller than current figures and add quantification graph.

3.      Current Figure 4: This important map should be shown as protein sequence numbers, because authors handle full and fragmented FlaA recombinant proteins. DNA size information should be transfer to Table 1. Amino acid numbers should be started from 1, not 0 (DNA also) and 511 at the C-teriminus end of full length of FlaA. Some of current DNA sizes in Figure 4 are not correct to translate to proteins, so please make sure and present both of protein size of fragmented FlaA (number of amino acid) and its GroEL T1-conjugated fusion protein size (predicted molecular sizes as kDa) in Figure 4. You can add additional bar of GroEL T1 alone fused to each of fragmented FlaA in this Figure. The most reactive Frag 2-5-2-2 should be shown its alignment of 17 amino acid sequences in this figure or in main text clearly.

4.      Serum samples of patients: Authors used serum from infant patients with (n=60) and without (n=221) H. pylori infection. Information of total 281 patients, such as sample numbers, mean and range of age, sex ratio, need to show as Table 2 in the first paragraph in Results section. H. pylori negative group and H. pylori positive group, and two sub-positive groups divided CLO test can be shown.

5.      Discussion: Add discussion about Frag 2-5-2-2 from aspects in 3D- structure of FlaA.

Minor points

1.      Page 5, lane 150: Authors described that the recombinant proteins were purified by glutathione-affinity chromatography using Glutathione Sepharose resin, according to the manufacture’s instruction (Invitrogen), although authors use pET15b vector based pEGexp vector to make fusion protein with GroEL T1. In this vector, purification may be designed to use His-tag originally. Please explain and modify or add the methods section.

Reviewer 3 Report

Summary

The manuscript by Park et al. describes an analysis of the antigenicity of H. pylori FlaA and fragments thereof, with the goal of identifying small FlaA fragments as diagnostic antigens for determining H. pylori infections via serology rather than by gastric biopsy. In general, the work is straightforward and generates promising results. I have a few comments for the authors’ consideration.

Major points.

Figures 2 and 3. The authors state (Fig. 2 legend) that GroEL T1 ‘did not shown any reactivity’ with the serum, however, in panel B there does appear to be reactivity. This is also seen in Fig. 3B, lane 1. So I would soften the statements of ‘no reactivity’ (also on lines 131 and lines 282-283) to reflect this.   

Figure 2. It would be helpful to indicate in the figure that fragments 2-1 to 2-6 are fusion proteins with GroEL T1 to make it more clear as to why fragments 2-1 to 2-6 appear as proteins larger than native fragment 2, from which those smaller fragments are derived. Additional text in the figure legend explaining this would also be helpful.

As a main finding of the work, the authors identified flaA fragment 2-5-2-2 as encoding a diagnostic antigen that can be used for serologic determination of H. pylori infection. As discussed by the authors, H. pylori strains are quite diverse both in virulence factor content as well as at the nucleotide level (especially geographically), the authors should provide some analysis of whether this particular peptide of H. pylori strain 51 FlaA (ISTVNNISITQVNVKAA) is highly conserved among H. pylori strains. Bacterial flagellins can have conserved and variable regions. If the peptide of interest is quite variable, it could limit the utility of this peptide for diagnosing more diverse H. pylori strains. Conversely, if it is highly conserved, it could greatly increase the usefulness of this approach. A multiple alignment of FlaA proteins from diverse H. pylori strains would address this directly.

Minor points.

Lines 103-104. This sentence should be reworded (‘which included NdeI …..regions of flaA’). It appears that the indicated nucleotide sequences are not in the flaA sequence but were added to generate the indicated restriction sites.

Legends for figures 5 and 6. Please define the asterisks shown in the figures as indicating P values.

Editorial / typographical.

Line 103. Typo in the restriction enzyme NdeI.

Line 110. Italicize flaA.

Lines 157-158. Please insert the verb ‘were’ between ‘recombinant proteins’ and ‘electrotransferred’ to give ‘recombinant proteins were electrotransferred’

Round 2

Reviewer 1 Report

The authors have done a good job responding to the reviewer comments. Two additional points remain:

1. Figure 5 shows that a number of controls have significant IgG reactivity against Frag 2252. What would account for this reactivity, and how does it affect the specificity of the test?

2. Line 270: It is unclear what "A flagellum" refers to. Do you mean flagellar protein?

Reviewer 2 Report

Authors improved Figure 1, Figure 2 and some parts of Figure 4. However, Figure 3 was replaced to less effective original western blotting as Figure 3B. Without good detection of western blotting image, quantification of blots can not be judged, I can not see Frag 2-5-2-2 is really positive from Figure 3B. The situation that authors exhausted all H. pylori-positive sera can not change the authors' responsibility to present result clearly. If authors have a small amount of positive sera remaining , even only Frag2-5-1-1, 2-5-2-1, 2-5-2-2 are good to do western blotting again. Lucking the most important key data, I can not keep to be positive for this paper.
